# A Simple yet Effective Adaptive Inter-organ Contrastive Learning Framework for Unsupervised Domain Adaptation

**Yiyou Sun**[1]                                                          YIYOUSUN@LINK.CUHK.EDU.HK
**Zheyao Gao**[2]                                                          ZHEYAOGAO@CUHK.EDU.HK
**Xiaogen Zhou**[2]                                                        XIAOGENZHOU@CUHK.EDU.HK
**Qi Dou**[2]                                                              QIDOU@CUHK.EDU.HK
**Winnie Chiu Wing Chu**[1]                                               WINNIECHU@CUHK.EDU.HK
[1] *Department of Imaging and Interventional Radiology, CU Lab of AI in Radiology, The Chinese University of Hong Kong, Hong Kong, SAR, China*

[2] *Department of Computer Science and Engineering, Institute of Medical Intelligence, The Chinese University of Hong Kong, SAR, China*

**Editors:** Under Review for MIDL 2026

## Abstract

Strong unsupervised domain adaptation (UDA) in multi-organ segmentation seeks to unify complementary information from heterogeneous imaging protocols within a single model without sacrificing source-modality performance, yet the substantial domain gap between modalities makes feature-level alignment non-trivial. Pseudo-label learning (PLL) has emerged as the dominant paradigm, but it suffers from information loss due to hard thresholding and bias introduced by class imbalance and noisy predictions. Contrastive learning (CL) offers a complementary direction by structuring semantic constrast, yet existing voxel-level formulations incur prohibitive computational costs on volumetric data and fail to capture the global anatomical context critical for organ segmentation. In this work, we propose Adaptive Inter-organ Contrastive Learning (AICL), a unified UDA framework for 3D multi-organ cross-modality segmentation that exploits PPL and CL synergistically to facilitate better cross-modality feature alignment. AICL employs dynamic soft pseudo-labels as guidance in the feature latent space to organize for inter-organ samples as positive-negative pairs for CL. Meanwhile, the model is trained with supervised consistency learning (SCL) using mixed ground truths and pseudo-labels, promoting a more discriminative and compact shared latent space. Extensive experiments and ablation studies on an orbital and a cardiac dataset reveal the effectiveness of each component and a significant advancement in segmentation results.

**Keywords:** Unsupervised Domain Adaptation, Multi-organ Segmentation

## 1. Introduction

Unsupervised domain adaptation (UDA) has become a cornerstone of cross-domain medical image segmentation (Qu et al., 2024; Lin et al., 2024c), transferring knowledge from a labeled source to an unlabeled target domain (Shin et al., 2023; Zhao et al., 2023; Zhou et al., 2025). The core challenge lies in the domain gap of substantial differences in intensity distributions, contrast profiles, and noise characteristics across modalities make direct feature-level comparison unreliable (Lee et al., 2021; Ma et al., 2026b; Zhang et al., 2026).

Early UDA methods emphasized either adversarial alignment in feature/output space or image-to-image translation. Adversarial alignment encourages global distribution matching

but can blur semantic boundaries and underfit minority structures (Hoffman et al., 2016; Chen et al., 2017; Vu et al.). Image translation (Park et al., 2020; Han et al., 2021) may partly reduce appearance gaps but risks altering anatomy and depends on cycle or structural constraints that are hard to satisfy in practice. Pseudo-label learning (PLL) (Zhao et al., 2023; Shin et al., 2023) has become the cornerstone of modern UDA in medical segmentation. By converting model predictions into supervision, teacher–student frameworks (Ma et al., 2026a; Chen et al., 2026) leverage uncertainty estimation (Lin et al., 2024a, 2025a, 2024b) and consistency checks to improve label quality. Despite these measures, inherent noise persists; hard thresholds lose information(Dumoulin et al., 2016; Wang et al., 2022), while class imbalance and low-confidence predictions introduce significant bias.

Contrastive learning (CL) offers a promising paradigm that addresses multi-organ segmentation via pulling consistent samples together while pushing apart dissimilar ones (Gu et al., 2024; Wang et al.; Zhang et al., 2023; Lin et al., 2025b). Most existing CL approaches, however, construct contrastive objectives at the *voxel level*, requiring exhaustive pairwise comparisons across full-resolution feature maps (Park et al., 2020). In 3D medical imaging, this voxel-wise paradigm relies on dense, voxel-level comparisons, resulting in a substantial computational bottleneck that limits scalability to high-resolution volumetric data. Moreover, pixel-to-pixel representations fail to capture the global contextual information essential for organ segmentation in medical images, where anatomical structures are inherently coherent.

Motivated by the above observations, we proposed adaptive inter-organ contrastive learning (AICL), a unified UDA framework for 3D multi-organ cross-modality segmentation that synergizes the semantic guidance of PLL with the representation power of CL in a structure-aware manner. Instead of discarding uncertain target predictions through rigid hard thresholding, AICL retains the soft semantic information encoded in pseudo-labels and exploits it to guide both organ-patch sampling and pseudo-label consistency regularization. In this way, pseudo-labels are not merely treated as noisy supervision, but are further leveraged as semantic cues to dynamically organize the latent space, encouraging tighter class-wise clustering and more robust cross-domain feature alignment. Furthermore, AICL selectively samples positive and negative pairs for CL and performs alignment at the feature-patch level, which is computationally prohibitive for volumetric data. This enables the model to focus on semantically meaningful organ representations while preserving richer local context than isolated pixel-to-pixel comparisons. Our contributions are summarized as follows: (1) We develop an organ-wise pseudo-label guided patch sampling (PGPS) strategy in the cross-modality feature alignment (CMFA) module for CL to guarantee optimal feature discrepancy and efficient feature representation. (2) We implement a pseudo-label-guided CL (PGCL) regularizer that complements the supervised consistency learning (SCL). This module effectively pushes apart cross-modality features from different classes while pulling together those from the same class in the latent embedding space, robust to the noise inherent in pseudo-labels.

## 2. Methods

The proposed adaptive inter-organ contrastive learning (AICL) framework processes interleaved inputs $I_m \in \mathbb{R}^{H \times W \times D}$ from source and target domains $m \in [m_1, m_2]$ within a unified

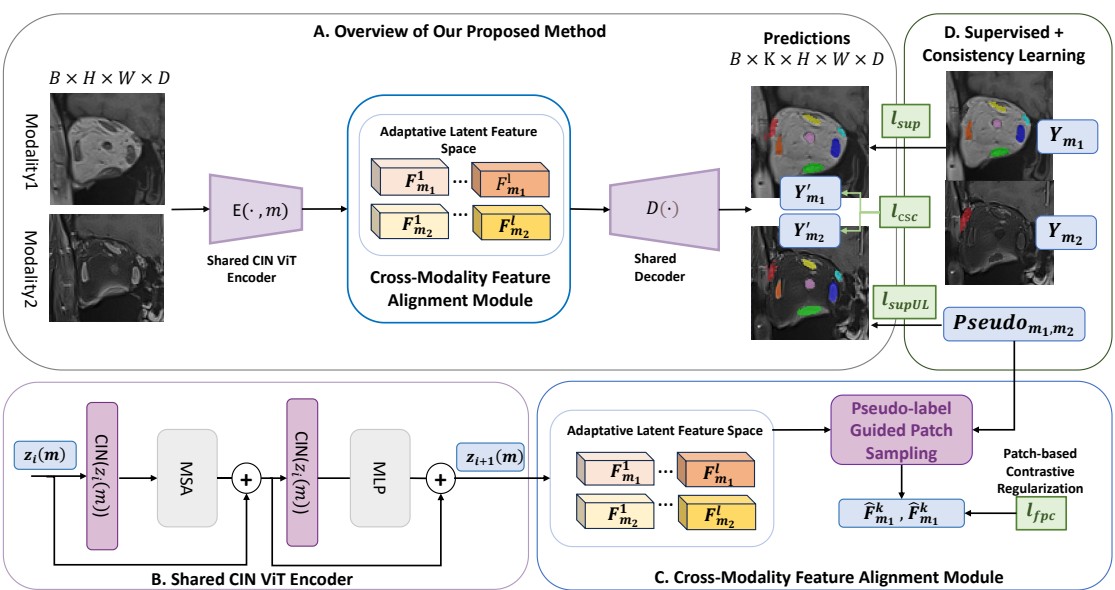

Figure 1: Flowchart of our proposed method. A. The overall pipeline takes interleaved multi-modal images as inputs. B. The framework utilizes a shared encoder to decouple modality-specific statistics from shared semantic representations $z$. C. CMFA module regularizes positive and negative paired samples generated from $F$ via PGPS by aligning cross-modal patch-wise features. D. SCL module jointly optimizes the model using ground truths $Y$ and pseudo labels $Pseudo$.

batch $B$ to generate comprehensive segmentation predictions. As illustrated in Fig. 1, our architecture first employs a shared conditional instance normalization (CIN) (Dumoulin et al., 2016; Bastico et al., 2023) vision transformer (ViT) (Guo et al., 2025) encoder to dynamically adapt feature distributions across diverse imaging modalities. Subsequently, CMFA systematically regularizes the shared latent feature space through PGCL, ensuring anatomical consistency between modalities. Finally, the framework integrates SCL to enforce consistency between predictions and corresponding available ground truth.

## 2.1. Modality-Adaptive Encoder

The shared-weight encoder $E(\cdot, m)$ integrates a CIN mechanism, parameterized by modality-specific scaling and shifting parameters $\gamma_m$ and $\beta_m$ for each modality input $I_1, I_2$. The encoder projects inputs into a unified yet modality-adaptive latent feature space $F_m \in \mathbb{R}^{C \times \frac{H}{4} \times \frac{W}{4} \times \frac{D}{4}}$ by independently normalizing instance-specific statistics across source and target domains. In each layer $l$ of latent feature space, feature maps $z_l(m)$ pass a learnable parameterized CIN which is defined as and shown in Fig. 1B:

$$CIN(z) = \gamma_m(\frac{z - \mu(z)}{\sigma(z)}) + \beta_m, \tag{1}$$

where $\mu(z)$ and $\sigma(z)$ represent channel-wise mean and standard deviation computed per instance within each batch.

Modality-specific learnable parameters $\gamma_m$ and $\beta_m$ are trained to decouple sensing-specific statistics (e.g., intensity, noise, and contrast) from modality-consistent semantic content, enabling a single shared encoder to generalize across modalities. CIN provides lightweight adaptation while preserving modality fidelity, thereby facilitating efficient training without compromising cross-modality alignment.

## 2.2. Cross-Modality Feature Alignment

### 2.2.1. Pseudo-label Guided Patch Sampling in Latent Feature Space

In our framework, PGPS is illustrated in Fig. 2. Specifically, inputs $I_m$ pass through a shared-weight encoder $E(\cdot, m)$ and a decoder $D$, yielding voxel-wise pseudo-labels $Y'_m = D(E(I_m, m)) \in \mathbb{R}^{K \times H \times W \times D}$. To perform cross-modality alignment in latent feature space, we downsampled $Y'_m$ to feature-layer resolution $\overline{Y'_m} \in \mathbb{R}^{K \times \frac{H}{4} \times \frac{W}{4} \times \frac{D}{4}}$ in spatial dimensions, where $k \in [1, 2, ..., K]$ indexes organ classes and $(h, w, d), h \in [0, \frac{H}{4}), w \in [0, \frac{W}{4}), d \in [0, \frac{D}{4})$ indexes voxel position. Leveraging the compactness of anatomical structures in medical images, we extracted organ-wise feature patch embeddings $\hat{F}_m^k \in \mathbb{R}^{K \times \hat{H} \times \hat{W} \times \hat{D}}$ from $F_m$. Specifically, for each class $k$, region of interest (ROI) centers $(h_c^k, w_c^k, d_c^k)$ are obtained as mean coordinates of $\overline{Y'_m}(k)$ valid voxels. We then crop a 3D cuboid $C_m^k$ around the center of a fixed patch size $ps = (H_{ps}, W_{ps}, D_{ps})$. Organ-wise feature patches $\hat{F}_m^k$ are sampled by bounding cubes in feature space to ensure they include a reasonable ratio of foreground and background semantic information. Relative equations are formulated as:

$$\mathcal{C}_m^k = \left\{ (h, w, d) \ \middle| \ |h - h_c^k| \leq r_h, \ |w - w_c^k| \leq r_w, \ |d - d_c^k| \leq r_d \right\}. \tag{2}$$

$$\hat{F}_m^k = F_m \big[ :, \ \mathcal{C}_m^k \big] \in \mathbb{R}^{C \times H_{ps} \times W_{ps} \times D_{ps}}. \tag{3}$$

where $(r_h, r_w, r_d)$ are half of $ps$ calculated by $2r_h + 1 = H_{ps}, 2r_w + 1 = W_{ps}, 2r_d + 1 = D_{ps}$.

For patch extraction complexity, dense pixel-wise contrastive loss operates on a full-resolution scale compared to our patch sampling strategy, that is $\mathcal{O}(HWD)^2$ compared to $\mathcal{O}(K \cdot ps^3)$ where K is the number of organ classes. Furthermore, the $ps$ is a hyperparameter empirically set according to organ size, balancing local details and global context. PGPS decouples the dependency on precise anatomical correspondence and preserves rich features exposed to the network, preparing robust multi-organ cross-modality feature alignment.

### 2.2.2. Pseudo-label Guided Contrastive Learning

Inspired by the PatchNCE (Park et al., 2020) that enforces local feature consistency by contrasting positive and negative image patch pairs, we propose PGCL to extend this concept to CMFA. PGCL leverages PGPS to create $\hat{F}$ when $l$ is settled as the last layer of $E(\cdot, m)$, and we compute the feature patch contrastive (FPC) loss as:

$$l_{fpc} = -\mathbb{E}_{m \sim (m_1, m_2)} \sum_{k=1}^{K} \log \left[ \frac{\exp(\phi(\hat{F}_m^k \cdot \hat{F}_m^k)/\tau)}{\exp(\phi(\hat{F}_m^k \cdot \hat{F}_m^k)/\tau) + \sum_{j=1, j \neq k}^{K} \exp(\phi(\hat{F}_m^k \cdot \hat{F}_m^j)/\tau)} \right], \tag{4}$$

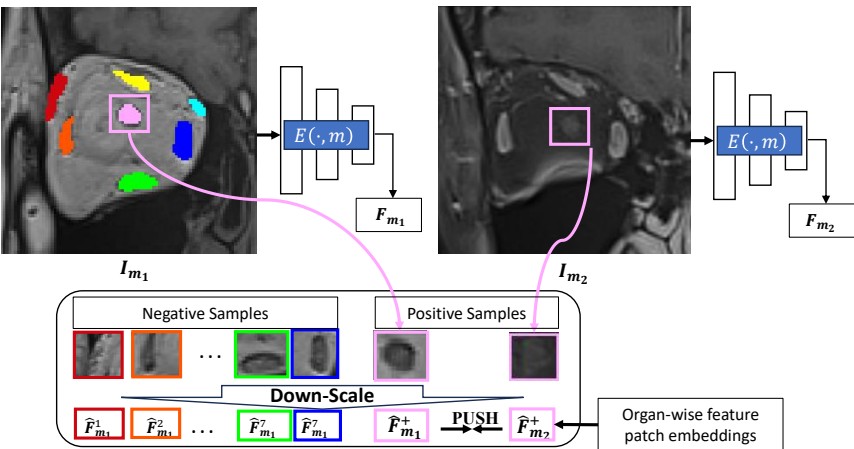

Figure 2: Illustration of PGPS. Pseudo labels are downsampled into the latent feature space for organ-wise feature embeddings $\hat{F}_m^k$ generation. The proposed method minimizes the distance ("PUSH") between positive pairs (same $k$), effectively facilitating cross-modality feature alignment.

where $\phi(\cdot)$ denotes cosine similarity, and $\tau$ is a temperature hyperparameter. Critically, positive pairs are feature patches sharing the same semantic class, while negative pairs come from different classes irrespective of modality.

PGCL enforces that same-class features are close, whether from different or same modalities, while different classes are separated at the organ-wise semantic feature patch level.

### 2.3. Supervised Consistency Learning

In the SCL module, $l_{sup}$ calculates the focal dice (Jadon, 2020; Lin et al., 2017) loss $l_{sup} = l_{dice} + l_{focal}$ between the predictions $Y'$ and ground truth $Y$ from annotated classes, while $l_{supUL}$ calculates the focal dice between $Y'$ and initial pseudo labels $Pseudo$ for organ regions without manual labels. To promote agreement between the image-scale features, contrastive structural consistency loss $l_{csc}$ enforces consistency between $Y'_{m_1}(k)$ and $Y'_{m_2}(k)$ by constructing positive and negative pairs based on shared or distinct anatomical structures (van den Oord DeepMind et al.). $l_{csc}$ is defined as:

$$l_{csc} = -\mathbb{E} \sum_{k=1}^{K} log \frac{\phi(Y_{m_1}^k, Y_{m2}^k)}{\sum_{j=1}^{K} \phi(Y_{m_1}^k, Y_{m2}^j)}, \tag{5}$$

Overall, the total loss of our framework is:

$$l_{total} = \lambda_{sup} l_{sup} + \lambda_{supUL} l_{supUL} + \lambda_{csc} l_{csc} + \lambda_{fpc} l_{fpc}, \tag{6}$$

where the $\lambda_{sup}, \lambda_{supUL}, \lambda_{csc}, \lambda_{fpc}$ are trade-off parameters scaling the importance of each loss component.

## 3. Experiments

### 3.1. Datasets

**TAO Dataset.** The in-house TAO dataset comprises 3D orbital MRI scans from 100 subjects, acquired through two complementary protocols: pre-contrast T1-weighted (T1) and post-contrast T1-weighted (T1c) imaging. The dataset contains full annotations for 20 cases, including extraocular muscle (EOM) groups, optic nerves (ON), and lacrimal glands (LG), partitioned into validation (20%) and test sets (80%). The remaining 80 training cases are partially annotated with EOM and ON on T1, and LG on T1c. A standardized preprocessing pipeline composed of image registration, cropping inputs to $96 \times 96 \times 32$ patches centered in regions with dense anatomical orbital structures, and normalizing intensity distribution to range $[0, 1]$. Subsequently, random 3D rotations and axis-aligned flips constitute data augmentation.

**MS-CMRSeg Dataset.** The publicly available MS-CMRSeg dataset (Zhuang, 2016, 2019) encompasses 45 paired cardiac imaging data. The protocol includes balanced-steady state free precession (bSSFP) cine sequences, serving as the source modality, and late gadolinium enhancement (LGE) sequences as the target modality. We only employ expert-validated annotations delineating three cardiac structures from bSSFP: the left ventricular cavity (LV), right ventricular cavity (RV), and left ventricular myocardium (Myo) across all cases. The dataset is randomly partitioned into 35 LGE/bSSFP for model training and 10 pairs for testing. Similar data processing steps as TAO are applied, and inputs are cropped into $480 \times 480 \times 16$.

### 3.2. Implementation Details

All experiments were conducted using Python 3.10 and PyTorch 1.13.1 on an NVIDIA A100 GPU with CUDA 11.7. We adopted SwinUNETR (Hatamizadeh et al., 2022) as the backbone architecture for TAO and UNET (Siddique et al., 2020; Tarvainen and Valpola) for MS-CMRSeg. SwinUNETR was configured with the feature size of $fs = 48$, encoder layer depth of $L = 4$, and hidden size of $K = 768$. To deal with a small-scale dataset, UNET comprises five resolution levels with feature widths $[16, 64, 128, 256, 512]$. In the encoder, each level begins with a residual unit for downsampling with strides of $[1, 2, 2, 2, 1]$. After iterative tuning of the hyperparameters from empirically initiated sets, we identified the optimal training protocol utilizing an Adam optimizer, learning rate is set as $1e-4$ with weight delay $1e-5$, $l_{sup} = 1$, and the overlap ratio is 0.5 for sliding window inference.

We use a ramp-up schedule for scaling factors $\lambda_{supUL}(t) = 1 - \min(1, \max(0, \frac{t-10}{T}))$, $\lambda_{csc}(t) = \min(1, \max(0, \frac{t-10}{T}))$, and $\lambda_{fpc} = 0.5 \min(1, \max(0, \frac{t-10}{T}))$ for more stable training where $T = 20$. Patch size is determined by a data-driven manner of statistically estimating organ size. Concretely, for each class, we compute the bounding-box extents of its ground-truth masks after downsampling to the latent feature resolution and summarize the extents across the training set using median and percentiles. We then select $ps = (7, 7, 7)$ for TAO and $ps = (25, 25, 9)$ for MS-CMRSeg to allow the masks $C_m^k$ to predominantly cover boundary regions for each organ while limiting background inclusion.

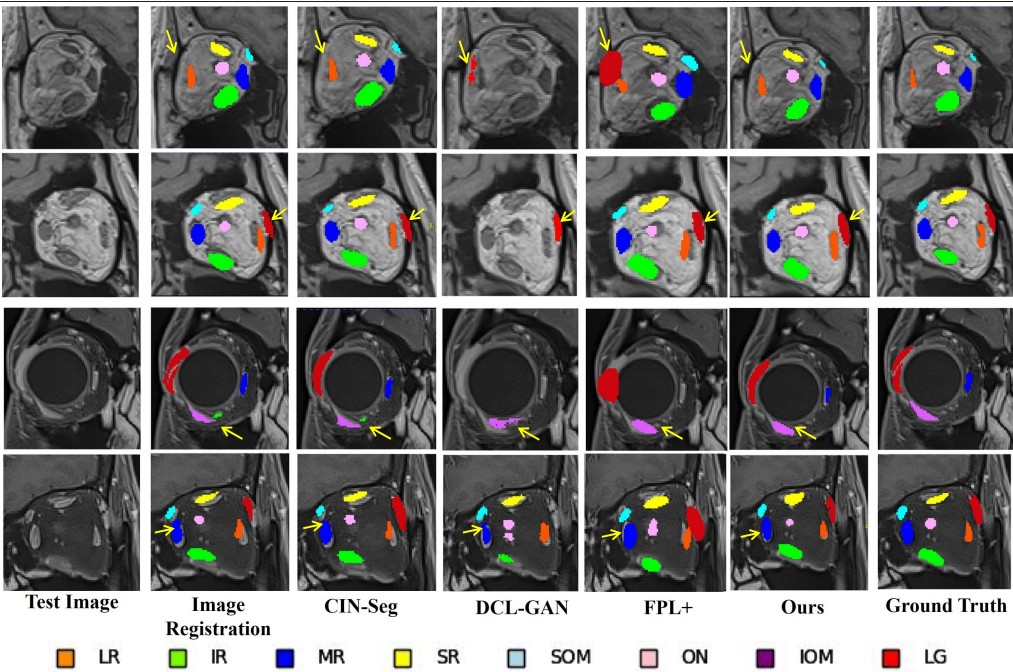

Figure 3: The qualitative comparisons of segmentation on the TAO dataset. Each row represents the same case, where top two rows come from T1 and bottom two rows come from T1c. Colors indicate different anatomical structures. Comparing EOMs and ON in T1c and LG on T1 (marked by yellow arrows) shows varying degrees of imprecision, particularly for small and low-contrast regions. In contrast, ours more consistently matches the ground-truth extent and location across rows, with cleaner boundaries and fewer false positives.

### 3.3. Effectiveness of Our Methods

We evaluated the effectiveness of our proposed method by measuring multi-organ segmentation performance in the Dice Similarity Score (Dice) and Hausdorff Distance 95 (HD95).

Table 1 presents quantitative comparisons of our proposed method against related unsupervised domain adaptation methods on the TAO dataset, adapting from T1 to T1c. REG compares the source domain labels transformed by a rigid matrix calculated from image registration across modalities and the target domain label. REG depicts the difficulty of resolving the domain gap between inputs. CIN-seg (Bastico et al., 2023) utilizes registered pseudo labels as supervision for cross-modal fusion, suffering an averaged 9.6% Dice decline over organs missing annotations compared to GT-supervised organs. Our method outperforms other methods by a large margin, especially in the missing label case, both in dice and HD95. DCLGAN(Han et al., 2021) leverages unsupervised contrastive learning for cross-modal image translation, enabling supervised learning on synthetic target data and source domain labels. However, anatomical distortions introduced during translation degrade performance (e.g., 28.9% Dice drop for LG in T1), as synthetic images often misalign

| Methods | LR | IR | MR | SR | SOM | ON | IOM | LG | **Avg.** |
|---------|-----|-----|-----|-----|-----|-----|-----|-----|-----|
| T1 Dice[%] ↑ | | | | | | | | | |
| REG | – | – | – | – | – | – | – | 61.31 | 61.31 |
| CIN-seg | 78.37 | 88.54 | 88.59 | 78.69 | 80.51 | 83.20 | 67.31 | 65.75 | 79.08 |
| DCLGAN | – | – | – | – | – | – | – | 55.40 | 55.40 |
| FPL+ | 55.12 | 71.64 | 70.13 | 66.07 | 56.95 | 49.69 | 16.20 | 60.50 | 55.79 |
| Ours | **82.60** | **90.31** | **89.74** | **80.76** | **82.89** | **85.57** | **72.08** | **68.24** | **81.52** |
| T1 HD95[mm] ↓ | | | | | | | | | |
| REG | – | – | – | – | – | – | – | 9.51 | 9.51 |
| CIN-seg | 6.13 | 3.41 | 3.48 | 8.70 | 9.54 | 7.16 | 6.47 | **7.47** | 6.54 |
| DCLGAN | – | – | – | – | – | – | – | 9.80 | 9.80 |
| FPL+ | 8.31 | 10.30 | 10.20 | 9.06 | 8.12 | 8.31 | 12.25 | 10.05 | 9.57 |
| Ours | **5.53** | **3.83** | **3.11** | **4.59** | **3.58** | **4.22** | **5.25** | 7.71 | **4.58** |
| T1c Dice[%] ↑ | | | | | | | | | |
| REG | 58.26 | 73.68 | 69.54 | 57.94 | 56.98 | 59.36 | 44.15 | - | 59.98 |
| CIN-seg | 70.50 | 78.21 | 79.40 | 68.38 | 70.06 | 68.68 | 51.06 | 77.12 | 70.43 |
| DCLGAN | 47.80 | 63.48 | 70.18 | 59.03 | 65.65 | 49.23 | 35.07 | – | 55.78 |
| FPL+ | 50.16 | 69.78 | 77.94 | 67.15 | 73.48 | 58.88 | 38.45 | 71.51 | 63.42 |
| Ours | **80.67** | **81.90** | **85.63** | **74.97** | **81.84** | **74.48** | **62.79** | **78.32** | **77.39** |
| T1c HD95[mm] ↓ | | | | | | | | | |
| REG | 6.83 | 5.05 | 7.02 | 5.38 | 7.00 | 5.23 | 9.25 | - | 6.53 |
| CIN-seg | 6.49 | 5.64 | 5.29 | 8.23 | 10.47 | 4.44 | 10.29 | 8.42 | 7.41 |
| DCLGAN | 20.13 | 13.32 | 9.93 | 10.50 | 10.03 | 20.94 | 14.12 | – | 14.14 |
| FPL+ | 5.71 | 4.93 | 5.20 | 5.57 | 3.48 | 5.51 | 10.53 | 9.27 | 6.28 |
| Ours | **5.98** | **4.68** | **4.17** | **5.05** | **2.88** | **4.34** | **8.97** | 7.71 | **5.47** |

Table 1: Comparison of different methods for the segmentation of TAO-affected organs on the T1 and T1c modality.

with ground truth structures. FPL+ (Wu et al., 2024) employs dual-domain pseudo-label generation with noise filtering. While effective in ideal scenarios, its reliance on heuristic thresholds amplifies error propagation under severe annotation sparsity. This framework performs better in T1c than T1 modality because it relies on synthetic T1c images with EOM and ON ground truth, and vice versa.

To intuitively verify the impact of the PGCL in CMFA module on feature representation learning, we visualized the distribution of feature embeddings $fp_m^k$ using t-SNE. Fig. 5A–B compare the feature spaces before and after applying PGCL. Feature distribution without PGCL exhibits loose distribution with low intra-class compactness. Specifically, IOM from T1 and ON from T1c show vague boundaries and potential overlap in the central region. This lack of distinct separability explains the baseline model's struggle with class discrepancy. In contrast, Fig. 5B demonstrates that introducing PGCL significantly regularizes

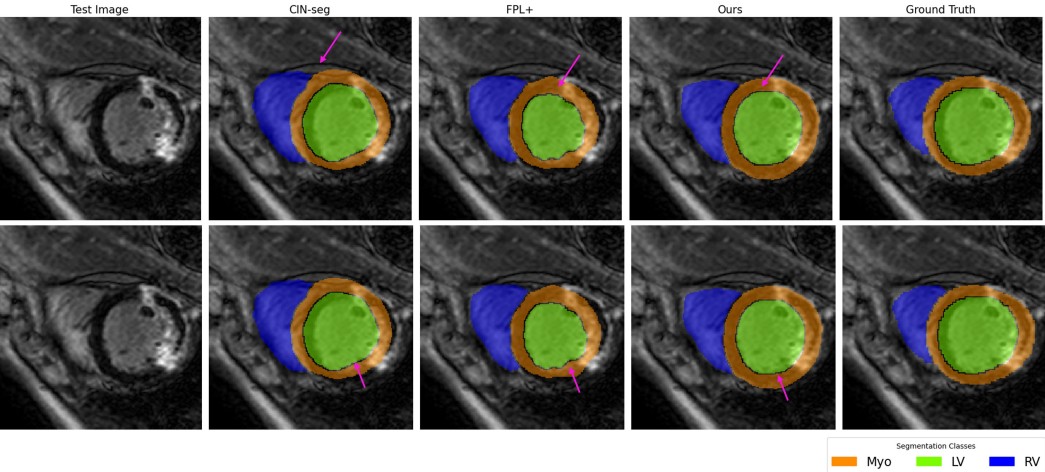

Figure 4: The qualitative results of segmentation on the MS-CMRSeg dataset. Different rows are from different cases.

the latent space. Enhanced semantic separability validates that our method effectively mitigates the domain shift problem by aligning feature distributions.

| Methods | Dice[%] ↑ | | | | HD95[mm] ↓ | | | |
|---|---|---|---|---|---|---|---|---|
| | Myo | LV | RV | Avg | Myo | LV | RV | Avg |
| REG | 56.99 | 79.74 | 73.27 | 70.00 | 10.22 | 9.45 | 27.73 | 15.80 |
| CIN-seg | 70.71 | 86.77 | **78.71** | 78.73 | 10.22 | 9.45 | 27.73 | 15.80 |
| FPL+ | 68.97 | 86.32 | 73.52 | 76.27 | 45.36 | 10.60 | 49.27 | 35.08 |
| Ours | **74.97** | **87.55** | 77.09 | **79.87** | **6.47** | **4.89** | **9.10** | **6.82** |

Table 2: Comparison of different methods for the segmentation on MS-CMRSeg adapting bSSFP to LGE.

### 3.4. Ablation Study

#### 3.4.1. EFFECTIVENESS OF KEY COMPONENTS

We conducted a systematic ablation study on the TAO dataset, focusing on the bidirectional domain adaptation between T1 and T1c modalities, to evaluate the contributions of the proposed core components, CMFA and SCL. As summarized in Tab. 3, these components are denoted as $l_{fpc}$, and $l_{csc}$, respectively. Our analysis began with a baseline model devoid of components, trained solely on cross-modal inputs with standard supervised dicefocal loss, achieving a modest average Dice of 69.05%, and a relatively high average HD95 of 9.57 mm. A closer inspection reveals significant performance bottlenecks in segmenting small, irregular structures and low-contrast boundaries of IOM, showing a particularly poor Dice

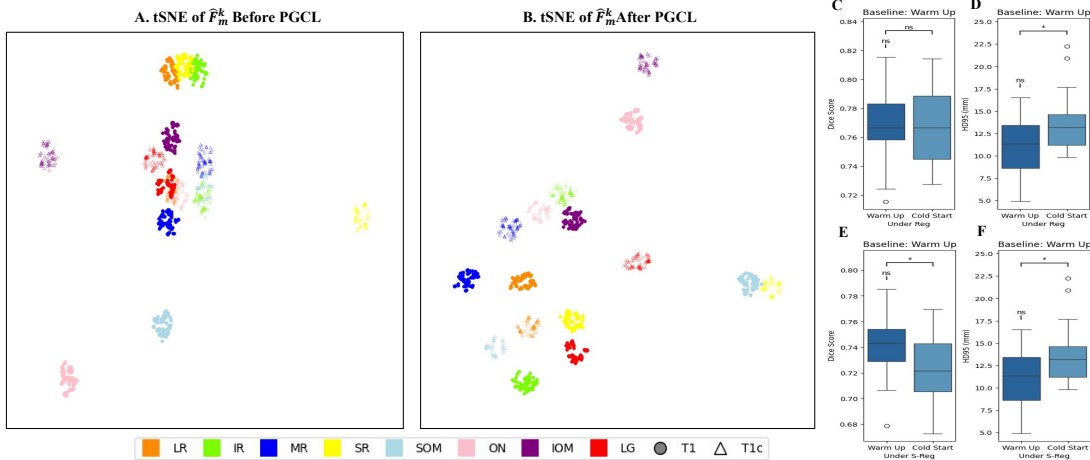

Figure 5: Left and right (A–B) t-SNE visualize feature embeddings from TAO dataset without and with PGCL applied. Different markers as $o$, $\triangle$ indicate features from T1 and T1c modality, respectively, with corresponding colors of organ groups marked in the legend. C–D shows Dice and HD95 comparison between training with 'Warm-up' and 'Cold-start' using REG as *Pseudo*. E–F uses Shifted REG.

score of 47.38%. This underscores the challenge of learning robust representations for fine-grained orbital structures under domain shift. The integration of both components achieves superior performance, validating the synergistic effect of our dual-consistency framework. The full model reaches an average Dice of 76.38% (a 7.33% improvement over the baseline) and drastically reduces the HD95 to 5.47 mm. Most remarkably, the segmentation of the challenging IOM, SOM improves by over 15%, 11% compared to the baseline. These results demonstrate that combining global alignment $l_{csc}$ with local feature refinement $l_{fpc}$ effectively mitigates domain discrepancies, ensuring anatomically plausible segmentation even for complex orbital structures.

### 3.4.2. Influence of Pseudo-label Quality

A potential concern arises that the model heavily relies on the quality of initial pseudo-labels to stabilize PGPG and PGCL, especially in the first several epochs of training. We deliberately corrupt the initialization by shifting the registered labels along a random axis denoted as S-Reg, creating extreme spatial misalignment. As shown in In Table 4, the shifted label transfer itself yields poor segmentation quality of S-Reg: Avg Dice 30.73%, Avg HD95 8.35 mm, confirming that the initialization is substantially degraded. Despite this, our method converges to strong performance Avg Dice 74.04%, Avg HD95 7.24 mm, substantially outperforming the CIN-seg Avg Dice 57.82%, Avg HD95 12.67 mm. This further proves our model robustly decouples the dependency on precise anatomical correspondence.

| $l_{fpc}$ | $l_{csc}$ | T1 → T1c | | | | | | | T1c → T1 | Avg |
|---|---|---|---|---|---|---|---|---|---|---|
| | | LR | IR | MR | SR | SOM | ON | IOM | LG | |
| | | | | | Dice[%] ↑ | | | | | |
| | | 72.15 | 76.82 | 81.47 | 69.61 | 70.46 | 68.72 | 47.38 | 65.75 | 69.05 |
| | ✓ | 79.55 | 81.89 | 81.75 | 72.41 | 77.27 | 69.28 | 44.22 | 67.15 | 71.78 |
| ✓ | | 80.20 | 80.32 | 84.10 | 74.88 | 76.74 | 70.84 | 53.87 | 66.27 | 73.40 |
| ✓ | ✓ | **80.67** | 81.90 | **85.64** | **74.97** | **81.84** | **74.48** | **62.79** | **68.24** | **76.38** |
| | | | | | HD95[mm] ↓ | | | | | |
| | | 8.31 | 10.30 | 10.20 | 9.06 | 8.12 | 8.31 | 12.25 | 10.05 | 9.57 |
| | ✓ | 7.79 | 7.09 | 7.20 | 5.90 | 7.10 | 5.29 | 9.68 | 7.69 | 7.21 |
| ✓ | | 7.42 | 8.27 | 5.45 | 5.96 | 5.09 | 5.14 | 9.13 | 9.87 | 7.04 |
| ✓ | ✓ | **5.98** | **4.68** | **4.17** | **5.05** | **2.88** | **4.34** | **8.97** | **7.71** | **5.47** |

Table 3: Ablation results of Dice and HD95 on unlabeled organs (LR, IR, MR, SR, SOM, ON, IOM in T1c, and LG in T1) from TAO dataset.

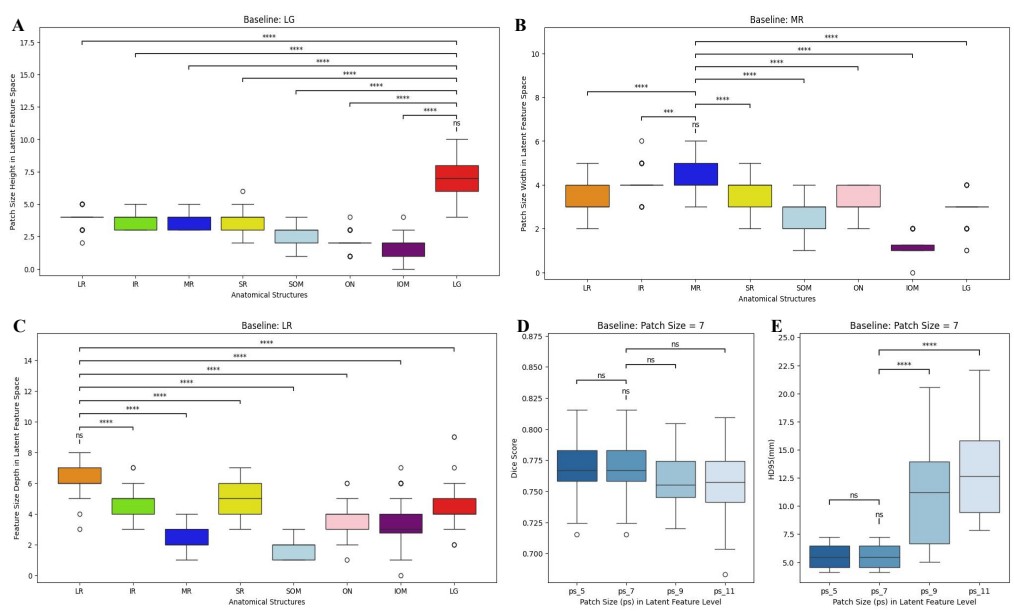

Figure 6: Organ-wise patch-size statistics and sensitivity analysis of PGPS. (A–C) Boxplots of the organ size extents of $H_{ps}, W_{ps}, D_{ps}$ respectively, measured in the latent feature space for each anatomical structure. (D–E) Robustness analysis of Dice score and HD95(mm) of the *isotropic* patch size $ps \in \{5, 7, 9, 11\}$ (applied at the latent feature resolution) for PGPS. The brackets indicate pairwise significantly differences where *ns*: not significant; *** $p < 0.001$; **** $p < 0.0001$.

We further study the comparsion of warm-up strategy and cold start under two initializations of registered labels (REG) and shifted registered labels (S-REG). As shown in Fig. 5 C–F, under S-REG initialization warm-up yields better enhancement of Dice and HD95 compared to REG initialization, indicating that delaying and ramping contrastive learning improves robustness when the initial pseudo labels are not reliable.

| Methods | T1 → T1c | | | | | | | T1c → T1 | Avg |
|---|---|---|---|---|---|---|---|---|---|
| | LR | IR | MR | SR | SOM | ON | IOM | LG | |
| Dice[%] ↑ | | | | | | | | | |
| S-Reg | 29.21 | 44.52 | 28.26 | 32.14 | 17.34 | 26.74 | 26.26 | 41.39 | 30.73 |
| CIN-seg | 71.47 | 68.10 | 64.63 | 51.64 | 52.38 | 61.84 | 40.81 | 51.72 | 57.82 |
| Ours | **78.54** | **79.14** | **85.50** | **72.85** | **79.72** | **71.41** | **63.19** | **61.98** | **74.04** |
| HD95[mm] ↓ | | | | | | | | | |
| S-Reg | 8.27 | 7.54 | **5.77** | 7.87 | 7.04 | **5.58** | 8.85 | 15.87 | 8.35 |
| CIN-seg | 8.81 | 8.18 | 8.26 | 9.34 | 9.69 | 6.65 | 12.47 | 37.98 | 12.67 |
| Ours | **7.37** | **6.50** | 6.77 | **7.22** | **6.43** | 7.19 | **8.84** | **7.59** | **7.24** |

Table 4: Robustness comparison under shifted registered labels (S-Reg) initialization under extreme spatial misalignment. Despite the substantial degradation in initialization quality, our method achieves markedly higher Dice and lower HD95.

### 3.4.3. Hyperparameter Analysis

We further analyze the sensitivity of $ps$ in PGPS. Fig.6A–C show that different anatomical structures occupy significantly different extents in the latent feature space. Fig.6D–E shows Dice score and HD95 robustness over varied *isotropic* patch size $ps \in \{5, 7, 9, 11\}$ at the latent feature level, where they map image patch size $I_{ps}$ ranges between $I_{ps} \in \{[16, 24), [24, 32), [32, 40), [40, 48)\}$. Contrastive regularization in feature space inherently allows the tolerance of a fixed patch size for all organ regions. In performance comparison, Dice remains statistically stable across the tested values, indicating that AICL is generally robust to moderate patch-size choices. However, HD95 degrades for larger patches $ps \geq 9$, suggesting that overly large cubes include excessive background and weaken the semantic purity of the sampled embeddings, which mainly harms boundary accuracy. A fixed $ps$ for all organs is primarily a practical constraint of our current PGPS strategy, where Organ-adaptive $ps$ with adaptive pooling or multi-scale patch sampling might be a nice improvement.

## 4. Conclusion

We propose a simple yet effective AICL method that combines SCL with CMFA normalized by CIN, to enhance alignment of cross-modality fine-grained semantic features. This strategy is advantageous for accommodating multi-modal data, by simply feeding interleaved inputs into the same batch. Our model performs comparably to or better than prevailing models in multi-organ segmentation from partly labeled multi-modal MRI.

## Acknowledgments

This work was supported in part by the General Research Funding from the Research Grants Council of the Hong Kong Special Administrative Region, China under Grant 14200721 and Grant 14100223 and in part by the Research Grants Council of the Hong Kong Special Administrative Region, China, under Grant T45-401/22-N, and in part by IdeaBooster Fund from Chinese University of Hong Kong University Grants Committee, under Grant IDBF25MED14.

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
