# OpenReview forum: "A Simple yet Effective Adaptive Inter-organ Contrastive Learning Framework for Unsupervised Domain Adaptation"
_MIDL.io/2026/Conference — MIDL 2026 Poster_

### Official Review · Reviewer_n8uE · 2025-12-22

**Confidence:** 4
**Preliminary Rating:** 4
**Final Rating:** 4

**Summary:**

The paper addresses the challenge of Unsupervised Domain Adaptation (UDA) in medical image segmentation, specifically focusing on the limitations of adversarial learning and standard contrastive learning (CL). The authors propose the Adaptive Inter-organ Contrastive Learning (AICL) framework. This framework utilizes a Modality-Adaptive Encoder with Conditional Instance Normalization (CIN) to handle multi-modal variations.

**Strengths:**

1. The PGPS strategy intelligently focuses on organ-relevant features via bounding cubes centered on pseudo-label voxels, effectively reducing computational bottlenecks associated with dense pixel-wise comparisons.
2. The use of Conditional Instance Normalization within a vision transformer (ViT) encoder provides a lightweight yet effective mechanism for decoupling sensing-specific statistics from semantic content.
3. By treating pseudo-labels as semantic cues rather than rigid supervision, the framework mitigates the impact of noisy labels and does not strictly rely on perfect anatomical correspondence between modalities.

**Weaknesses:**

1. The selection of the patch size for PGPS is empirically set based on organ size, which may limit the framework's out-of-the-box generalizability to new anatomical regions without manual tuning.
2. While the method aims to exploit the intrinsic structure of pseudo-labels, it still fundamentally relies on the initial quality of these labels for sampling, which could lead to reinforcement of early segmentation errors.

**Detailed Comments:**

1. The reliance on empirically set patch sizes according to organ size is a significant manual overhead. In a multi-organ setting with vastly different organ scales (e.g., the small optic nerve vs. larger muscles in the TAO dataset), a static or manually tuned patch size may be suboptimal for some structures.
2. While the paper claims PGPS avoids computational bottlenecks, it does not provide a formal comparison of training time or memory consumption (FLOPs/Parameters) against the dense pixel-wise comparisons it seeks to replace.

**Justification Of Final Rating:**

Thank you for the detailed response and the effort put into the rebuttal. While my concerns are now mostly resolved, I believe the current score still accurately reflects the paper's value. I will keep my original rating.

**Justification Of The Preliminary Rating:**

The paper maintains a clear and logical structure that progresses from a well-defined abstract through detailed methodological and experimental sections. It is driven by a strong motivation to address the limitations of adversarial learning and sparse feature spaces in cross-modal medical segmentation.

**Questions To Address In The Rebuttal:**

See weakness and details comments.

---

> ### Author Response · Authors · 2026-01-25
>
> ### Weakness1 and Comments1:
>
> Thanks for raising this important point regarding the potential manual overhead and limited out-of-the-box generalizability caused by human patch-size selection.
>
> **Automatic selection of patch size**
> To address this limitation, we have implemented an automatic and dataset-adaptive patch size selection strategy. For a new dataset, $ps$ can be computed directly from the training-set size statistics (median/percentiles of downsampled GT extents in feature resolution), requiring no manual search and selecting a more robust patch size. The rationale to choose patch size is to make sure feature patches contain predominantly organ contours (to provide informative, discriminative features) while avoiding excessive background, which may dilute the semantic signal and introduce noise into positive/negative pairs.  Details are supplemented in Section3.2.
>
> **How is model performed in multi-organ setting for a manually tuned patch size**
> Compared to automatic selection, $ps=5$ that falls near the average size of majority organs and $ps=7$ that falls near the largest size among all organ statistics, manually tuned patch size is not accurate enough to find the optimal $ps$ for overall organ segmentation.  The performance of different $ps$ is compared in Section 3.3.3.
>
> We additionally acknowledge that using a single fixed $ps$ for all organs can still be suboptimal for specific structures in extreme multi-scale settings. This is primarily a practical constraint of our current patch-wise contrastive implementation, which compares/batches fixed-size feature tensors. We now explicitly state this limitation and discuss future extensions (e.g., class-adaptive $ps$ with resizing/adaptive pooling or multi-scale patch sampling).
>
> ### Weakness2:
> Thanks for raising this important concern. We agree noisy pseudo labels is a critical challenge, and we implement initialization of registered labels and warm-up optimization.
>
> **Model reliance on initial pseudo label quality**
> To address potential early error enhancement during training, we initialized registered labels from the other modality instead of purely model-generated pseudo-labels at epoch 0, which provides a substantially better (more stable and better boundaries of regions) starting point.
> To cope with reliance of PGPS and PGPS on pseudo labels, we reformulate $\lambda_{fpc},\lambda_{csc}$ as ramp-up scaling factors for more stable training. By comparing the performance decrease of warm-up against cold-start setting when using different qualities of initial pseudo-labels, we find that this warm-up strategy plays a crucial role in preventing early pseudo-label noise from dominating optimization, especially when initial pseudo-labels are not reliable.
>
> **Model Robustness analysis under degraded pseudo-labels**
> To directly evaluate the risk of “reinforcing early errors,” we added a robustness experiment in Section3.3.2 where we intentionally degrade the initial pseudo labels by shifting the registered labels. Even when the initialization quality is very poor Avg Dice $0.307$, our method still converges to strong performance Avg Dice $0.740$, significantly higher than CIN-seg Avg Dice $0.578$. This proves our approach does not heavily rely on precise initial cross-modality correspondence.
>
>
> ### Comments2:
> Thanks very much for noting this omission.  We agree that the computational advantage should be supported more explicitly. Since PGPS/PGPL are implemented as loss-side modules, and they do not change the segmentation backbone architecture; therefore, the **model parameters and forward-pass FLOPs remain unchanged**.
> The additional cost arises only from (a) patch extraction and (b) contrastive similarity computation. For patch extraction, instead of enumerating all foreground voxels (e.g., `nonzero`) or performing dense pixel-wise matching, we obtain and only store valid patch anchors using **3D boolean projections**. For complexity of contrastive similarity, dense pixel-wise contrastive loss operates on a full-resolution scale compared to our patch sampling strategy, that is $\mathcal{O}(HWD)^2$ compared to $\mathcal{O}(K \cdot ps^3)$ where K is the number of organ classes.

---

### Official Review · Reviewer_LMzp · 2026-01-08

**Confidence:** 3
**Preliminary Rating:** 4

**Summary:**

The authors propose an unsupervised domain adaptation framework using contrastive learning that leverages pseudo-labels for organ-wise feature patch sampling. The authors compare their methods to other state-of-the-art approaches, perform ablation studies of the loss function, and utilize two distinct datasets. However, some improvements could be made to the qualitative results figures, as well as the tables, by including additional metrics and a statistical analysis.

**Strengths:**

1. The authors provide a well-written introduction with sufficient background, motivation, and clear contributions.
2. The authors clearly describe their method and architecture.
3. The authors use two datasets of different anatomy for training and validating their methods.
4. They also provide a comprehensive ablation study for the components of the loss function for both datasets.

**Weaknesses:**

1. In the definition of the loss function (equation 6), there seems to be no weighting factors for the l_sup component (Dice loss and focal loss).
2. The figures describing the results (3,4,5) could use some improvement.
3. The tables could use additional metrics and a statistical analysis.
4. The manuscript would benefit from a discussion on limitations and future work.

**Detailed Comments:**

Extended comments:
1. In the definition of the loss function (equation 6), there seems to be no weighting factors for the l_sup component (Dice loss and focal loss). Could the authors please comment on this and explain why a weighting factor was not needed?
2. In Figure 3, please explain what each row is, and provide a legend for the colors. Additionally, in the caption, include some observations of the segmentation performance. Guide the user to errors in the segmentation by using bounding boxes or arrows. In Figure 4, it is very hard to see the differences between the two markers; perhaps use larger markers or more distinctive shapes (circle and cross, for example). For Figure 5, it would be good to zoom in and include arrows for any errors/observations. Again, please include a legend for the colors. For both figures, in each row, it seems like the same crop was not utilized, as slight shifts in the images are visible.
3. In Table 1, the standard deviation of the Dice and HD95 values should be included. Additionally, please note that the top row represents structures. It would also be useful to include statistics to indicate if the results from your method are statistically different from the other state-of-the-art methods. Please also provide further details about how the other state-of-the-art methods were chosen and trained. Table 3 should also include the standard deviation. For the cardiac dataset, why not compare other state-of-the-art methods?
4. Please include limitations and future work in your conclusion.

Minor comments:
1. Why were two different backbone architectures used? (SwinUNETR for TAO and UNET for MS-CMRSeg).
2. Please correct the spelling mistakes (Haudorff → Hausdorff, and multi-oragn → multi-organ).
3. There are a lot of abbreviations that can be confusing for the reader to keep track of. Therefore, I recommend that the authors avoid abbreviations at least in the section headings.

**Justification Of The Preliminary Rating:**

The manuscript has a number of strengths, including clear contributions, a well-written introduction and background, and comprehensive experiments. However, there could be some improvements in the figures to improve the clarity, as well as improvements to the tables, and an inclusion of limitations and future work.

**Questions To Address In The Rebuttal:**

1. In the definition of the loss function (equation 6), there seems to be no weighting factors for the l_sup component (Dice loss and focal loss). Could the authors please comment on this and explain why a weighting factor was not needed?
2. Could the authors please include limitations and future work?
3. For the tables, please perform a statistical analysis to determine if results from your method are different from the other methods. If you could please also include how the other methods were chosen and trained that would be useful.

---

> ### Author Response · Authors · 2026-01-25
>
> ### Question 1 and Comment1 :
> Thanks very much for noting this omission. We have revised the manuscript to include weighting term of $\lambda_{sup}$. In our framework, $\lambda_{sup}$ controls our primary anchor $l_{sup}$ that directly optimizes predictions with available ground truths,  whereas $l_{csc}$ and $l_{fpc}$ act as regularization terms for unstable feature alignment that can otherwise dominate training if not property controlled and tuned. For this reason, we intentionally keep the supervised component dominant and set the weighting on the supervised terms to unit weight $1$ (i.e., $\lambda_{sup}=1$ and internal weights for balancing dice and focal are also fixed to 1).
> ### Question 2:
> We appreciate this suggestion and agree that the manuscript will benefit from a clearer discussion of limitations and future directions. In the revised version, we supplemented several sections, tables and charts to analyze the model dependence of pseudo-label quality, cold-start or use warm-up schedule for training, sensitivity analysis of patch size. As future work, we will highlight that adaptive organ-wise patch sizing (rather than one fixed patch size for all organs) is a promising direction, particularly for multi-scale organ segmentation.
>
> ### Question 3 and Comment 3:
> Thanks for this valuable recommendation. We strengthen the quantitative presentation using boxplots with p-value annotations in appendix as statistics to compare to other SOTA methods. (We note that due to formatting constraints, placing full per-organ mean$\pm$std for all settings directly in the main table may exceed the allowed table width).
>
> **Baseline choice and training details**
> We compared our methods within the scope of unsupervised domain adaptation methods, where DCLGAN is a representative image-level translation method which encourages global distribution matching but can blur semantic boundaries and underfit minority structures. We included FPL+ as a comparison for it refines pseudo labels as soft labels to avoid the common pitfall of treating unlabeled structures as background. This provides a meaningful comparison to our approach, since it contrasts direct pseudo-label updating with pseudo-label guided patch sampling (PGPS) and contrastive learning (PGCL) strategy design. Furthermore, CIN-seg uses registered labels as pseudo labels to supervised the cross-modality segmentation, offers a meaningful comparison with our updated pseudo labels guided strategies.
> ### Minor Comment 1:
> Thank you for raising this point. We used different backbones due to dataset characteristics and overfitting considerations. The MS-CMRSeg dataset contains only 35 training samples; in our preliminary experiments, SwinUNETR tended to overfit in this low-data regime, whereas U-Net provided more stable training and better generalization. For the TAO dataset, SwinUNETR consistently outperformed U-Net, and therefore we adopted it as the backbone for that dataset.
>
> ### Comments 2 and other minor comments:
> We greatly appreciate these careful observations of figure improvements, typos and abbreviation reduction on title of section. We have thoroughly checked and polished them.

---

### Official Review · Reviewer_dev5 · 2026-01-09

**Confidence:** 4
**Preliminary Rating:** 4

**Summary:**

The paper proposes Adaptive Inter-organ Contrastive Learning for unsupervised domain adaptation in medical segmentation. Addressing limitations of adversarial and pixel-wise approaches, it introduces Cross-Modality Feature Alignment using pseudo-labels to guide organ-specific patch sampling for contrastive learning. A modality-adaptive encoder with Conditional Instance Normalization further mitigates global shifts. The method outperforms state-of-the-art baselines on orbital and cardiac MRI datasets.

**Strengths:**

1. The proposal to shift from pixel-to-pixel contrastive learning to an "organ-wise feature patch" approach is scientifically sound and innovative. By using bounding cubes derived from pseudo-labels ($fp_m^k$) to define semantic correspondence, the method cleverly circumvents the strict spatial alignment required by standard methods, which is often violated in cross-modality medical imaging.
2. The integration of Conditional Instance Normalization within the Vision Transformer encoder is a novel architectural choice. It effectively decouples modality-specific style statistics (intensity, contrast) from shared semantic content without requiring heavy separate encoders, enhancing the efficiency of the adaptation process.
3. The proposed method show good results, particularly on the internal TAO dataset involving small, intricate structures like Extraocular Muscles and Optic Nerves. The ablation studies in Tables 2 and 3 clearly validate the individual contributions of the contrastive feature loss ($l_{fpc}$) and consistency loss ($l_{csc}$).

**Weaknesses:**

1. The core contribution, PGPS, relies entirely on the quality of pseudo-labels ($Y'_m$) to generate the bounding boxes for patch sampling. The paper does not adequately address the "cold start" problem. At the beginning of training, pseudo-labels are likely noisy or empty for the target domain. If the model fails to detect an organ initially, the mask $MS_m$ would be empty, potentially destabilizing the contrastive loss or leading to the collapse of that specific class representation. The robustness of this bootstrapping process is not analyzed.
2. The patch size ($PS$) for sampling is described as a hyperparameter "empirically set according to organ size". This reliance on manual, organ-specific tuning limits the method's generalization and "unsupervised" nature. It implies that for every new dataset or organ, a user must manually determine the optimal bounding box size, which is practically cumbersome. A sensitivity analysis regarding $PS$ is missing.
3. The mathematical formulation, particularly Equation 3 regarding the bounding box indices, appears complex and potentially contains typos (e.g., mixing notations like $M_{km}^*$ and $m_{k}$). Furthermore, the distinction between the "feature patch contrastive loss" ($l_{fpc}$) and the "contrastive structural consistency loss" ($l_{csc}$) could be clearer; specifically, how negative pairs are sampled for $l_{csc}$ in the output space versus the latent space is somewhat ambiguous in the text.

**Detailed Comments:**

1. In Section 3.3, "multi-oragn" should be corrected to "multi-organ".
2. In Section 3.1, the description of cardiac structures could be cleaner. "Left ventricular myocardium (Myo)" is standard, but consistent capitalization would help.
3. In Figure 3, maybe highlighting the specific regions where AICL improves over FPL+ (e.g., with arrows or zoomed-in crops) would make the qualitative advantage more obvious to the reader.

**Justification Of The Preliminary Rating:**

The paper presents a solid contribution to the field of medical UDA. The idea of "organ-wise" contrastive learning via patch sampling is novel and addresses valid limitations of pixel-wise approaches. The results, especially on the challenging orbital dataset, are persuasive. I have rated this as a "Weak Accept" rather than "Strong Accept" primarily due to the heuristic nature of the patch sizing (requiring manual tuning per organ) and the lack of analysis regarding the "cold start" stability of the pseudo-label guidance. Addressing the sensitivity and stability questions in the rebuttal would strengthen the submission.

**Questions To Address In The Rebuttal:**

1. How sensitive is the model performance to the choice of the patch size ($PS$)? Did you perform an experiment where $PS$ was fixed or varied? What happens if $PS$ is too large (including too much background)?
2, How does the PGPS module behave during the first few epochs when pseudo-labels are noisy? Did you implement a "warm-up" period where the contrastive loss is disabled until pseudo-labels stabilize?

---

> ### Author Response · Authors · 2026-01-24
>
> ### Weakness1 and Question 2:
> Thanks for bringing up this very important matter. We agreed that the empty or noisy pseudo-labels in first steps will not only destabilize the contrastive loss but also add difficulty for the model to converge.
>
> **How do we avoid empty pseudo-labels at beginning**
> In our case, the pseudo-labels are initialized using rigid transformation applied to ground truth from the other modality. Registered labels avoid empty maps in target domain by providing rough boundaries and locations of organs. In this way, initial pseudo labels offer supervision in design of supervised consistency learning (SCL) to strengthen the model stability during first epochs. Clearer descriptions of this initialization are added in Section 2.4 and Fig.1.
>
> **How does PGPS module behave during the first few epochs when pseudo-labels are noisy**
> To exploit the PGPS performance on noisy pseudo-label, we conduct a sensitivity study in Section 3.3.2 that intentionally creates a “worst-case” initialization via shifting registered labels (S-Reg) 3-pixel along any axis. This also mimics the worse misalignment across modalities contributing to worse initialization for us.
> **Warm-up Implementation**
> We agree cold-start is a critical challenge and warm-up is important, and we implement a warm-up period to check if our model’s performance will differ under different initial pseudo-label settings. Specifically, $\lambda_{fpc}, \lambda_{csc}$ are ramp-up scaling factors, and $l_{supUL}$ for supervision using pseudo labels is a ramp-down scaling factor between epoch 10 and 20.
>
> We add warm-up strategy experiments in Section 3.3.2 and observe that warm-up provides larger gains when the initialization is challenging. Specifically, under the S-REG, the average Dice across all TAO organs for the corrupted pseudo labels themselves, our model with cold start, and our model with warm-up are $0.307, 0.721, 0.744$ respectively, indicating that bootstrapping  remains stable even under extreme misalignment, and that the warm-up scheduling further mitigates early-noise effects by preventing PGPS/PGCL from over-committing to unreliable early pseudo labels.
>
> ### Weakness2 and Question 1:
> Thanks very much for raising this important concern regarding manual tuning of the patch size.
>
> **Patch Size was fixed or varied?**
> We acknowledge that manual tuning of $ps$ can reduce the framework’s out-of-the-box generalizability to new datasets and anatomical targets. We set a fixed patch size ahead of training using a data-driven strategy that coordinates the core concept of our PGPS and PGPL design and select a more-robust ps than before. This strategy is illustrated supplementally in Section 3.2. This procedure uses only the available training annotations (as standard in UDA/source-supervised training) and does not require any target-domain labels.
>
> **Sensitivity analysis of patch size**
> We then analyzed the sensitivity of patch size in TAO by setting $ps=[5,7,9, 11]$ where $ps=5$ comes from  averaged patch size across all organs, $ps=7$ comes from largest patch size across all organs, $ps=9, 11$ exceed largest patch size (including more background information, ps=9 is former choice). The key observation is that performance is robust to moderate patch-size changes (Dice remains statistically similar across tested values), while overly large patches degrade boundary accuracy (HD95 increases), consistent with the concern that large cubes include more background and dilute organ-boundary features used in contrastive pairing. This supports our choice of patch sizes that primarily capture organ boundaries while limiting background inclusion.
> ### Weakness3:
> Thanks a lot for noticing the confusing mathematical formulation in section 2.2.1 (especially for equation 3). The bounding boxes are cropped from original feature maps (from modality 1 and 2) as samples for constructing positive and negative pairs of contrastive loss. In this process, indices are used. Specifically, we use $Y, F$ to represent all scales (in spatial dimension) of pseudo-labels (mask) and feature maps, respectively. To distinguish them, we added different subscripts such as $\hat{Y}, \overline{Y}$. Cropping using indices in equation 3 is easier-to-understand in this version. We also clarify $l_{csc}$ created negative samples directly from prediction of full-image resolution when $l_{fpc}$ works in feature resolution level. We have thoroughly checked our notations and polished mathematical formation in our revised manuscript.
> ### Detailed Comments:
> Thanks for your kind detailed comments about typos and unclear visual comparison of segmentation figures, they are all corrected.

---

### Author Rebuttal · Authors · 2026-01-25

**Rebuttal:**

We sincerely thank the program chairs and the reviewers for their time, careful reading of our manuscript, and constructive feedback. We appreciate the positive assessment of our evaluation and the thoughtful suggestions on how the paper can be further strengthened. Below is  our revised manuscript.

**Supporting Material:**

/attachment/d6dee1ee9a4ef3402674224e16ea1e5750079d82.pdf

---

### Comment · Area_Chair_wvCD · 2026-02-01
**Please enter Final Ratings**

Dear reviewers,
Please note that today, Feb 1 is the last day to enter your final ratings. Thank you to those who have already updated. If you have not yet, please take a moment to look through the author’s rebuttal and update your final score and reasoning.
We greatly appreciate your important contribution to MIDL.
Thank you!
Your AC

---

### Meta-Review · Area_Chair_wvCD · 2026-02-09

**Recommendation:** Accept (Poster)
**Confidence:** 5

**Metareview:**

Reviewers all agree that this paper meets the bar for acceptance. They note strengths of the proposed method of organ-wise contrastive learning for unsupervised domain adaptation and good experimental results. Similar concerns raised by reviewers (regarding reliance/selection of patch size, pseudo-labels) were considered addressed by the rebuttal, which included new experiments and analysis, and additional clarifications were addressed. I agree with the reviewers' overall assessments and recommend acceptance of this paper.

---

### Decision · Program_Chairs · 2026-02-13

Accept (Poster)